# Residual learning for 3D motion corrected quantitative MRI: Robust clinical T1, T2 and proton density mapping

**Carolin M. Pirkl**[1,2]                                    CAROLIN.PIRKL@TUM.DE
**Matteo Cencini**[3,4]                                     MATTEO.CENCINI@GMAIL.COM
**Jan W. Kurzawski**[5]                                     JAN.KURZAWSKI@GMAIL.COM
**Diana Waldmannstetter**[1]                               DIANA.WALDMANNSTETTER@TUM.DE
**Hongwei Li**[1,6]                                         HONGWEI.LI@TUM.DE
**Anjany Sekuboyina**[1,7]                                  ANJANY.SEKUBOYINA@TUM.DE
**Sebastian Endt**[1,2]                                     SEBASTIAN.ENDT@TUM.DE
**Luca Peretti**[8,3,4]                                     LUCAPERETTI.LP@GMAIL.COM
**Graziella Donatelli**[9,4]                                GRAZIELLA_DONATELLI@HOTMAIL.COM
**Rosa Pasquariello**[3]                                    ROSA.PASQUARIELLO@FSM.UNIPI.IT
**Mauro Costagli**[3,10]                                    MAURO.COSTAGLI@FSM.UNIPI.IT
**Guido Buonincontri**[3,4]                                 GUIDO.BUONINCONTRI@GMAIL.COM
**Michela Tosetti**[3,4]                                    MICHELA.TOSETTI@FSM.UNIPI.IT
**Marion I. Menzel**[*2,11]                                 MENZEL@GE.COM
**Bjoern H. Menze**[*1,6]                                   BJOERN.MENZE@UZH.CH

[1] *Department of Computer Science, Technical University of Munich, Garching, Germany*

[2] *GE Healthcare, Munich, Germany*

[3] *IRCCS Fondazione Stella Maris, Pisa, Italy*

[4] *Fondazione Imago7, Pisa, Italy*

[5] *Pisa Division, National Institute for Nuclear Physics (INFN), Pisa, Italy*

[6] *Department of Quantitative Biomedicine, University of Zurich, Zurich, Switzerland*

[7] *Department of Neuroradiology, Klinikum rechts der Isar, Munich, Germany*

[8] *Department of Computer Science, University of Pisa, Pisa, Italy*

[9] *Azienda Ospedaliero-Universitaria Pisana, Pisa Italy*

[10] *Department of Neuroscience, Rehabilitation, Ophtalmology, Genetics, Maternal and Child Sciences (DINOGMI), University of Genova, Genova, Italy*

[11] *Department of Physics, Technical University of Munich, Garching, Germany*

## Abstract

Subject motion is one of the major challenges in clinical routine MR imaging. Despite ongoing research, motion correction has remained a complex problem without a universal solution. In advanced quantitative MR techniques, such as MR Fingerprinting, motion does not only affect a single image, like in single-contrast MRI, but disrupts the entire temporal evolution of the magnetization and causes parameter quantification errors due to a mismatch between the acquired and simulated signals. In this work, we present a deep learning-empowered retrospective motion correction for rapid 3D whole-brain multiparametric MRI based on Quantitative Transient-state Imaging (QTI). We propose a patch-based 3D multiscale convolutional neural network (CNN) that learns the residual error, i.e. after initial navigator-based correction, between motion-affected quantitative

---

[*] Contributed equally

T1, T2 and proton density maps and their motion-free counterparts. For efficient model training despite limited data availability, we propose a physics-informed simulation to apply continuous motion-patterns to motion-free data. We evaluate the performance of the residual CNN on 1.5T and 3T MRI data of ten healthy volunteers. We analyze the generalizability of the model when applied to real clinical cases, including pediatric and adult patients with large brain lesions. Our study demonstrates that image quality can be significantly improved after correcting for subject motion. This has important implications in clinical setups where large amounts of motion affected data must be discarded.

**Keywords:** 3D multiparametric MRI, motion correction, deep learning, residual learning, multiscale CNN

## 1. Introduction

Motion robustness is a key feature for routine imaging in general. It is especially crucial for pediatric or elderly patients and for patients affected by diseases that prevent them from maintaining a still position throughout the acquisition. It is therefore a clinical priority to develop techniques that effectively resolve motion artifacts. As their appearance highly depends on the individual acquisition, e.g. the used readout schemes, the targeted clinical question, the condition of the patient and the body region to be imaged, there is no universal solution. Consequently, a number of conceptionally different correction methods have been presented, ranging from prospective to retrospective, image-based methods (Zaitsev et al., 2015; Godenschweger et al., 2016).

Fast 3D multiparametric MRI techniques based on transient-state MRI (Ma et al., 2018; Gómez et al., 2020) are excellent candidates for the clinical practice, as they offer high quantification accuracy together with high repeatability and reproducibility (Buonincontri et al., 2021). Their reduced scan times enable improved motion robustness compared to conventional quantitative MRI with lengthy scanning protocols. While motion artifacts are generally reduced in these fast acquisition schemes, they are not entirely immune to motion. In fact, subject movements do not only affect a single time point of the acquisition, but corrupt the entire temporal magnetization evolution, captured by the acquired k-t space, and therewith subsequent parameter estimation. While previous work on motion correction for transient-state imaging has mainly concentrated on 2D acquisition schemes (Mehta et al., 2018; Cruz et al., 2019; Xu et al., 2019), there is only little work on motion correction for 3D multiparametric MRI.

Kurzawski et al. (2020) presented a navigator-based retrospective rigid motion correction for a 3D Quantitative Transient-state Imaging (QTI) technique based on a segmented readout scheme to acquire the k-t-space. Their proposed motion correction strategy relied upon self-navigators embedded within each acquisition segment, which enabled the recovery of a critical amount of the underlying parameter information degraded due to subject motion occurring between consecutive segments. Despite significant improvement of the image quality, resulting quantitative T1, T2 and proton density (PD) maps showed remaining artifacts originating from subject movements on a time-scale below the temporal resolution of the self-navigators of 7 s.

Here, we propose a deep learning (DL) method to resolve artifacts due to continuous motion that are not captured by the navigator-based approach. Our work is motivated by recent advances of DL at the interface between MR physics and medical computer vision

that have been demonstrated to make MR imaging more robust to subject motion (Usman et al., 2020; Oksuz, 2021; Gong et al., 2021; Pawar et al., 2018; Miao et al., 2016; Hou et al., 2018a,b), e.g. by directly removing motion-induced artifacts or by estimating the underlying motion parameters for subsequent realignment. We adopt the concept of residual learning (Zhang et al., 2017; Jin et al., 2017; Liu et al., 2020) and propose a 3D multiscale residual convolutional neural network (CNN) to improve on the previously presented navigator-based motion correction, presetting the following contributions: (1) We propose a 3D multiscale residual CNN to learn the non-linear relationship between the motion-corrupted T1, T2 and PD maps and the residual error maps, i.e. the deviation from the motion-free counterpart that remained after navigator-based correction (Kurzawski et al., 2020). (2) We rely on a 3D CNN architecture that captures the intrinsically 3D nature of the subject movements together with the 3D MR acquisition scheme to efficiently resolve motion artifacts and infer high-quality T1, T2 and PD maps. (3) We present a physics-informed simulation framework to retrospectively apply realistic continuous motion patterns to motion-free datasets, enabling a supervised training setup without the necessity for large amounts of paired acquisitions or fully sampled data. (4) We evaluate the performance of the proposed method on 1.5T and 3T MRI data of ten healthy volunteers who underwent QTI imaging twice: the first time they kept their head as still as possible, and the second time they voluntarily moved their heads during acquisition. We also apply our method to clinical cases, including pediatric and adult patients with large brain lesions, to demonstrate its generalizability and capability to improve motion-affected datasets in cases with pathological findings.

## 2. Material and methods

### 2.1. Residual learning for retrospective 3D motion correction

We propose a residual learning technique to resolve artifacts that could not be corrected by the navigator-based method of Kurzawski et al. (2020), which is recapped below to present a more complete picture. We demonstrate our method with its key components, the residual CNN model and the physics-informed motion simulation, on data acquired with 3D QTI.

**Navigator-based rigid motion correction**  The navigator-based correction identifies motion-induced misalignment in the acquired image-time series. To do so, the full k-t-space data is subdivided into subsequently acquired segments, from which we reconstruct equal-contrast navigator images. These navigators are then aligned to the first baseline navigator to estimate the spatial mismatch and to subsequently correct the k-t-space data accordingly. The corrected k-t-space data is then fed into the reconstruction pipeline as described in 2.2 to yield the motion-corrected parametric maps. The massive spatial undersampling of the fast 3D acquisition scheme limits the resolvable motion time-scale to 7 s as the lower SNR in temporally higher resolved self-navigators hampers a correct realignment.

**Residual learning CNN architecture and training**  We propose a 3D patch-based multiscale residual CNN to learn the deviation of the motion-corrupted parameter maps from the high-quality, motion-free reference. Learning a residual mapping has been shown to be more effective than a direct mapping (Zhang et al., 2017; Tamada et al., 2019; Ulas et al., 2018) as the residual maps capture a more sparse representations of the artifacts.

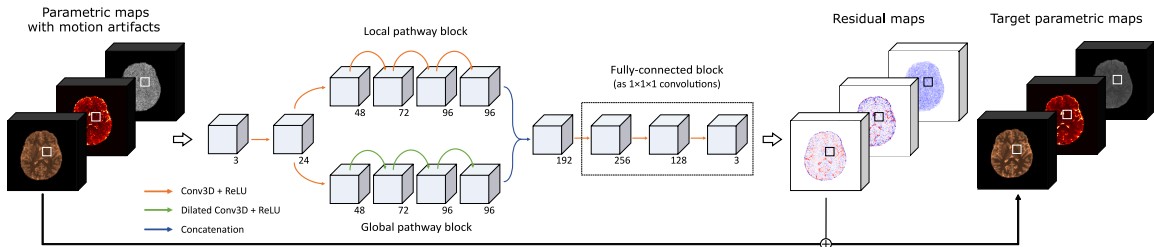

Figure 1: The multiscale CNN receives the parametric maps after navigator-based correction as input and outputs the residual maps.

The proposed CNN architecture[1] (Figure 1) receives a 3D input patch ($24 \times 24 \times 24$ voxels) of the quantitative maps degraded by motion artifacts that remained after navigator-based correction. The quantitative T1, T2 and PD parameters are reflected by three input channels. The model then spans out in a local and a global pathway. This dual pathway structure was shown to efficiently account for spatial image information on multiple scales (Kamnitsas et al., 2017; Kim et al., 2017; Ulas et al., 2018): The local path (with 3D convolutions and ReLU activations) processes more localized, spatially adjacent features. The dilated 3D convolutional layers in the global path allow to gather more global, contextual information due to an increased receptive field (Table A.2). Local and global features are concatenated and fed into a block of fully-connected layers, efficiently processing the decoded spatial relationships. To maintain the spatial dimensions throughout the network, the fully-connected layers are implemented as convolutional layers with $1 \times 1 \times 1$ kernels, to eventually output the residual maps, i.e. the difference of the navigator-corrected and the motion-free maps. We trained the residual CNN based on in-vivo 3D QTI data from ten healthy volunteers. For each subject, two datasets were acquired with the instruction to hold still for the first scan and to rapidly move the head during the second session as detailed by Kurzawski et al. (2020). All subjects were scanned on a 1.5T and a 3T scanner (HDxt and MR750 scanners, GE Healthcare, Milwaukee, WI) with the sequence parameters described in 2.2. For a supervised training setup, we only considered the motion-free data and created a database of artificially motion-corrupted 3D QTI data as described below. The in-vivo data with real motion was only used for testing. The DL-model was then trained to learn the residual maps between the parametric maps with simulated motion artifacts and the motion-free counterpart. The retrospectively corrupted data of seven subjects were used for model training and two subjects' data for validation, with $10,000/3,000$ randomly sampled 3D patches, respectively. The remaining subject data was held back for testing. We trained the residual CNN for a maximum of 100 epochs with a batch size of 20, using Adam optimization to minimize the L1 loss function with a learning rate of $1e^{-4}$, keeping the model state with the best validation loss.

**Physics-informed simulation of motion-corrupted data**    To allow the proposed DL-model to learn how diverse motion patterns propagate to the inferred multiparametric

---

1. Code available on https://github.com/CarolinMA/MRP_MoCo

maps, we simulated motion-corrupted data from the motion-free 3D QTI acquisitions. To do so, we applied continuous rigid motion patterns, i.e. translation and rotation, to the individual time frames of the acquired k-t-space. To imitate continuous head movements, we continuously varied the misalignment of consecutive k-t-space time points. We achieved a realistic artifact appearance as we applied ranges of the artificial translation and rotation patterns as experimentally observed by Kurzawski et al. (2020), i.e. translations $-20\,\mathrm{mm} \leq \Delta x, \Delta y, \Delta z \leq 20\,\mathrm{mm}$ and rotations $-20° \leq \Delta roll, \Delta pitch, \Delta yaw \leq 20°$. We then performed a navigator-based correction to mitigate artifacts due to inter-segment movements in the first place. The thereby obtained parametric maps with remaining artifacts due to intra-segment movements, illustrated in Figure A.1, were the input to the CNN.

## 2.2. Data acquisition and processing

**In-vivo data**  All in-vivo data presented in this study were acquired in accordance with the 1964 Helsinki declaration and its later amendments or comparable ethical standards. Approval was granted by the local ethics boards.

**MR acquisition and reconstruction**  In-vivo data from ten healthy volunteers, a pediatric and an adult patient were scanned with an inversion-prepared 3D SSFP QTI implementation (Gómez et al., 2020) with variable flip angle ramps, TI=18 ms, TE=0.5 ms and TR=8.5 ms. The acquisition of transient state image series relies on in-plane spirals with spherical rotations to sample the k-t-space (=3D+time, i.e. $225 \times 225 \times 225\mathrm{mm}^3$ field of view with $1.125 \times 1.125 \times 1.125\mathrm{mm}^3$ isotropic voxel size and 880 time points). By design, the acquisition is built by consecutive segments (n=56) of the same excitation scheme, each with a duration of 7 s, and iteratively fills the k-t-space by randomly permuting in-plane and spherical rotation angles of the readouts. The k-t-space data are then reconstructed using zero-filling, followed by projection onto a low rank subspace, gridding onto a Cartesian grid, 3D inverse fast Fourier transform and subsequent coil sensitivity estimation and combination. Quantitative maps of T1, T2 and PD are estimated by matching the reconstructed subspace images to a pre-computed dictionary with granularity and parameter ranges as specified in Kurzawski et al. (2020).

## 2.3. Experimental setup

**Cross-validation experiment on healthy volunteer data**  We evaluated the performance of the residual CNN, trained on solely simulated motion, in a ten-fold cross-validation experiment by repeating the training setup, as described in 2.1, ten times. Following this leave-one-out scheme, the data of the held-back volunteer with real motion after initial navigator-based correction was used for model testing in each instance. At test time, we divided the parametric maps into 3D patches of $24 \times 24 \times 24$ voxels, shifted along all three dimensions with a step size of 4 voxels, for patch-wise CNN processing. Predicted residual error patches are added to the motion-corrupted input and averaged to eventually yield the full 3D motion-corrected T1, T2 and PD maps. We ran the cross-validation experiment for 1.5T and 3T data individually. The obtained quantitative maps were compared to the co-registered motion-free reference using the voxel-wise concordance correlation coefficient (CCC) and coefficient of determination ($R^2$) as performance metrics.

**Generalization analysis on clinical quantitative MRI** For further performance analysis, we applied the best-performing model in the cross-validation experiment to clinical 3D QTI scans of a pediatric (8-year old) patient with subtotal agenesis of the corpus callosum, scanned at 1.5T, and an adult patient with glioblastoma, scanned at 3T.

## 3. Results and discussion

**Cross-validation experiment on healthy volunteer data** The proposed 3D residual CNN, trained on purely artificially corrupted data, provided T1, T2 and PD maps with an image quality comparable to the motion-free reference maps. This is observed when visually comparing the quantitative maps of a representative test case of the cross-validation experiment for both the 1.5T (Figure 2(a), Figure A.2) and 3T data (Figure 2(b), Figure A.3).

Quantitative evaluation of the cross-validation experiment by means of the CCC and $R^2$ (Table A.1) substantiates the qualitative finding and reflects the improvement achieved by the navigator-based realignment and the subsequent residual CNN. For both 1.5T and 3T data, quantitative measures indicate that the residual CNN further improved the outcome of the navigator-based correction for all parametric maps. As already visually observed, mean CCC and $R^2$ values reflect the higher impact of the DL-model on T2 and PD than T1 maps. Furthermore, Table A.1 shows that after CNN-based motion-correction, we achieved better correspondence with the motion-free reference for the 3T data than for the 1.5T scans. However, from Figure 2(a) and Figure A.2 we observe that the residual CNN does not only remove motion-induced artifacts, but additionally suppresses noise-like aliasing. This effect is more pronounced for the 1.5T data with intrinsically lower SNR and image quality than for a 3T field-strength with higher SNR. The additional denoising results in parametric maps with image qualities that go beyond the motion-free reference, which in turn explains the lower overall agreement observed with the motion-free reference acquisitions.

The cross-validation experiment shows that the combination of the residual CNN with the navigator-based correction efficiently resolves head movements on two time-scales: 1) The self-navigator-based estimation and subsequent realignment of motion-induced displacements in the k-t-space has proven to recover a significant amount of the parameter information corrupted by abrupt inter-segment movements. 2) With the 3D residual multi-scale CNN, we reliably resolve residual artifacts and phase inconsistencies due to continuous intra-segment movements that are unresolved by the limited temporal resolution of the self-navigators, providing high-quality and artifact-free parameter maps.

The proposed physics-informed motion simulation allows us to retrospectively apply continuous motion directly to the k-t-space and propagate it through the reconstruction pipeline. We make implicit use of the forward encoding operator from k-space to parameter-space to generate self-contained, paired training data for supervised model training. Thus, we present an efficient training scheme that does not require large amounts of motion and motion-free data pairs to be acquired. Also, in contrast to other physics-guided methods, we do not rely on fully sampled data to be used as reference for supervised network training. This is from particular practical relevance as the acquisition of fully-sampled 3D+time QTI data is infeasible due to prohibitively long scan times (Yaman et al., 2020).

The 3D patch-based CNN implementation allowed us to fully capture the spatial correlations that inevitably arise from 1) the subject movements in the 3D space, which cause

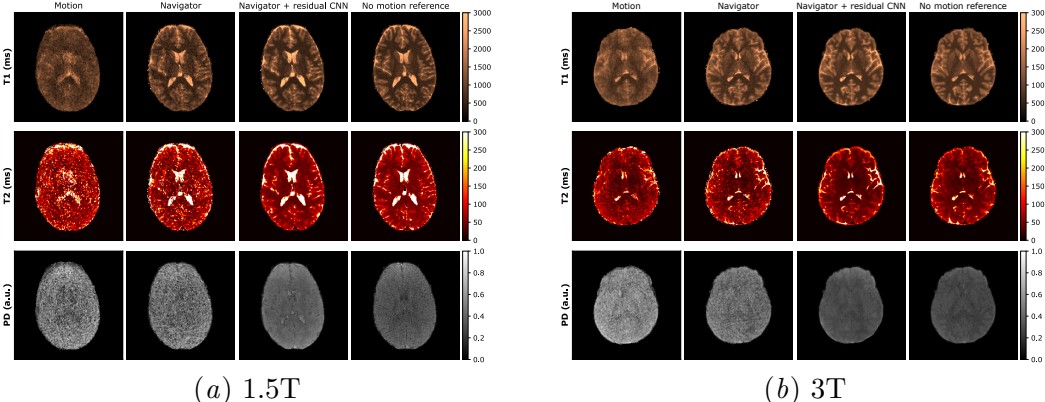

$(a)$ 1.5T $\qquad\qquad\qquad$ $(b)$ 3T

Figure 2: Proposed motion correction for representative volunteer scans at 1.5T (a) and 3T (b) (axial views). T1, T2 and PD maps show pronounced motion-induced artifacts (*Motion*) compared to the *No motion reference*. Remaining artifacts after *Navigator*-based correction are resolved by the residual CNN (*Navigator + residual CNN*), providing high-quality parameter maps.

spatially correlated image artifacts, and 2) the 3D design of the MR acquisition with spatial undersampling and multicoil imaging that provoke a mixing of signal components. With the adaption of the residual learning concept, we transferred the non-linear disentangling of the primary parameter information and the secondary image artifacts into the sparse representation of the residual maps.

**Generalization analysis on clinical quantitative MRI**  For the clinical test cases at 1.5T and 3T, Figure 3, Figure A.4 and Figure A.5 indicate that the residual CNN yields high-quality, artifact-free parametric maps. In both cases, the navigator-based approach did not improve image quality of the parametric maps as much as seen for the volunteer data (Figure 2, Figure A.2, Figure A.3). This is attributed to the fact that there were no pronounced abrupt movements but the patients moved their heads continuously, i.e. on a faster scale of what can be resolved by the self-navigators. The patient datasets also showcase the generalization capabilities of the residual CNN. We observed reliable motion-correction results in the presence of pathological findings in both adult and pediatric patients whose brain anatomy differs from that of healthy adults in the training data.

**Limitations and outlook**  Although the proposed multiscale CNN has shown convincing efficiency and functionality in this proof of concept, more advanced DL architectures might have the potential to improve on our baseline. We also plan to further investigate on the intrinsic denoising capacities of our method as revealed by the 1.5T experiments. As suggested from the clinical evaluation, patient data seemed to be affected by continuous head movements without any abrupt position changes. It is hence subject to our current and follow-up work to investigate what motion scales can be resolved by the residual CNN when applied as a stand-alone tool. We also plan to explore potential application scenarios of the presented DL-empowered motion correction in other body regions and motion patterns.

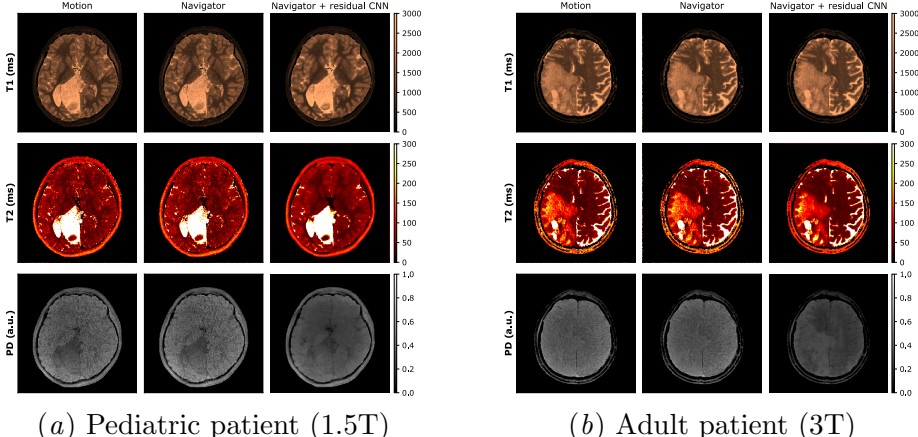

($a$) Pediatric patient (1.5T)  ($b$) Adult patient (3T)

Figure 3: Proposed motion correction for representative clinical test cases (axial views). (a) Pediatric patient with subtotal agenesis of the corpus callosum and interhemispheric cyst, scanned at 1.5T. (b) Adult patient with glioblastoma in the temporo-parietal region with cystic-necrotic and hemorrhagic components, and marked perilesional edema, scanned at 3T. For both patients, the residual CNN improved image quality of all parametric maps (*Navigator + residual CNN*), mitigating image artifacts attributed to head movements during scan sessions.

## 4. Conclusion

In this work, we propose a 3D multiscale residual CNN for retrospective motion correction in fast 3D whole-brain multiparametric MRI. We present a physics-informed motion simulation, allowing for efficient model training without the requirement of large amounts of paired data. The 3D CNN architecture captures the intrinsically 3D relationships of the motion-induced corruptions to reliably recover high-quality T1, T2 and PD maps. Taking advantage of the sparsity in the residual maps, we can substantially improve the quality of quantitative maps suffering from subject movement - in case of healthy volunteers but also for pediatric and adult patients with pathological findings. This is particularly important in clinical setups where scans frequently have to be repeated, possibly under sedation, because of motion artifacts. With fast scanning time and higher motion-immunity, quantitative MRI may become a standard for clinical practices.

## Acknowledgments

This project receives financial support from the Italian Ministry of Health and the Tuscany Region under the project Ricerca Finalizzata, grant No. GR-2016-02361693, Deutsche Forschungsgemeinschaft (DFG) through Research Training Group GRK 2274, TUM International Graduate School of Science and Engineering (IGSSE), GSC 81, and the European Union's Horizon 2020 research and innovation programme, grant agreement No. 952172.

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

## Appendix A.  Supplementary figures and tables

### A.1.  Physics-informed simulation of motion-corrupted data

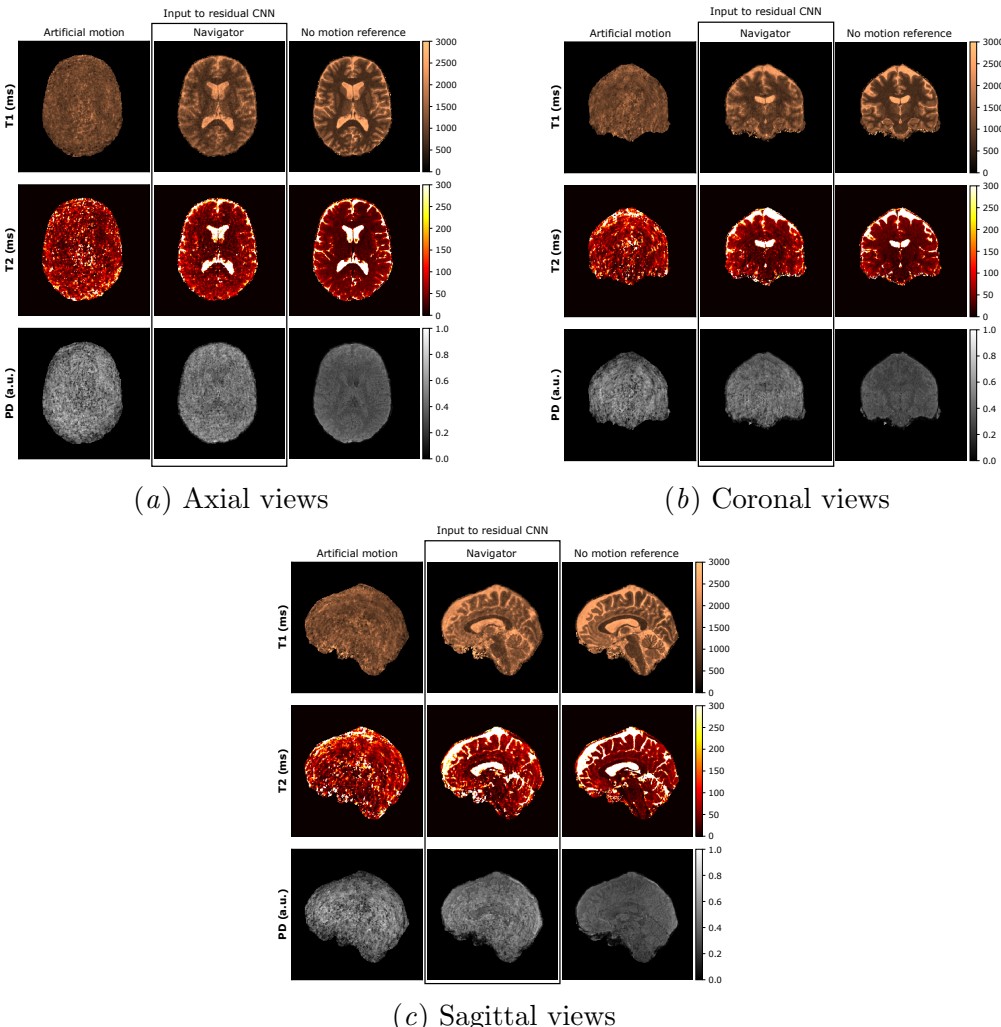

Figure A.1: Physics-informed motion simulation illustrated for a representative volunteer dataset acquired at 3T. Continuous rigid, i.e. translation and rotation, motion patterns are applied to the individual time frames of the acquired motion-free k-t-space data (*No motion reference*), imitating continuous head movements (*Artificial motion*). Navigator-based motion correction is then applied to mitigate artifacts due to inter-segment movements in the first place (*Navigator*). The obtained parametric maps with remaining artifacts due to continuous intra-segment movements are the input to the residual CNN.

## A.2. Cross-validation experiment on healthy volunteer data

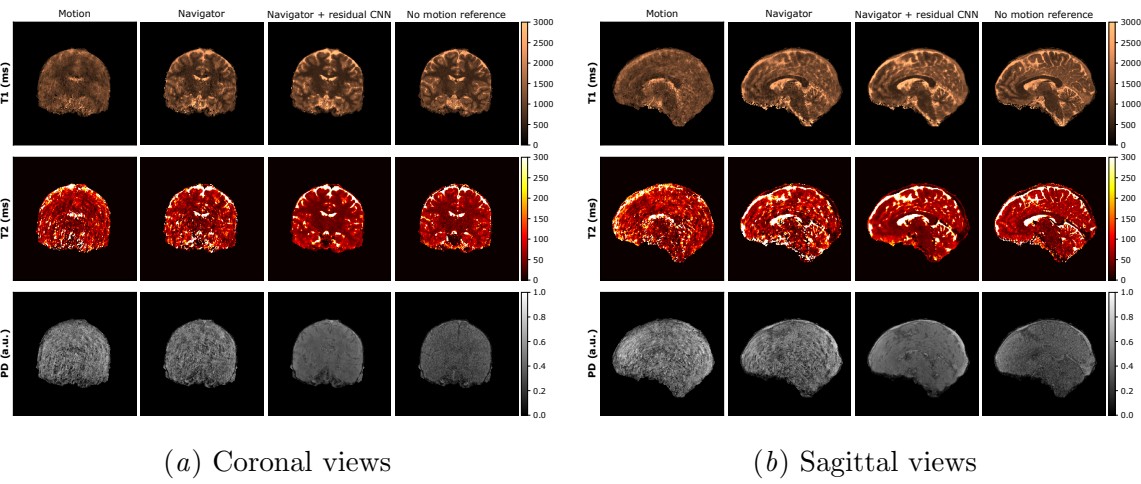

(*a*) Coronal views        (*b*) Sagittal views

Figure A.2: Proposed motion correction for a representative volunteer test dataset acquired at 1.5T (coronal and sagittal views).

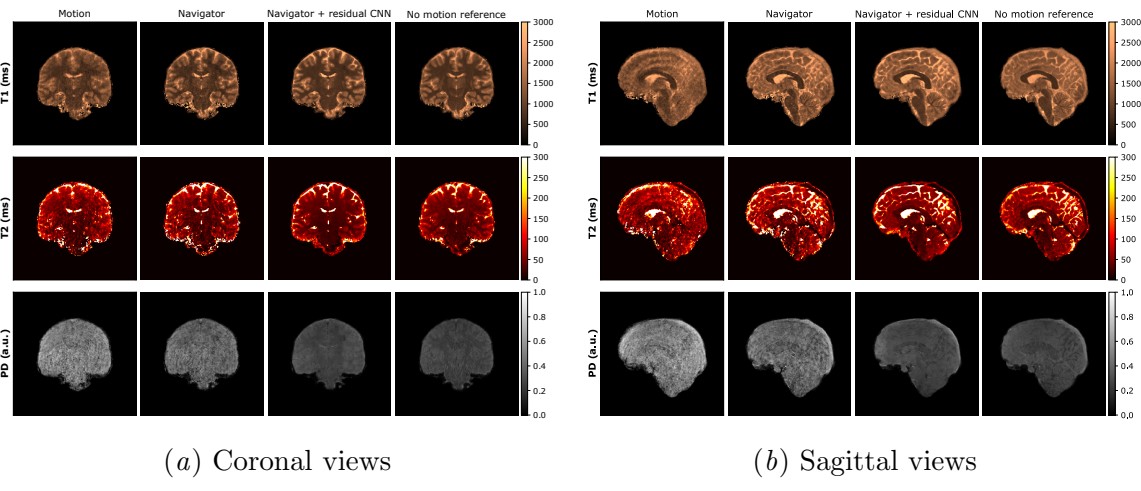

(*a*) Coronal views        (*b*) Sagittal views

Figure A.3: Proposed motion correction for a representative volunteer test dataset acquired at 3T (coronal and sagittal views).

Table A.1: Quantitative evaluation of the cross-validation experiment for motion-corrupted, measured volunteer data summarized by concordance correlation coefficient (CCC) and coefficient of determination ($R^2$) metrics between the result of the respective correction method, i.e. only navigator-based correction (*Navigator*) and navigator-based correction with subsequent residual CNN-based correction (*Navigator + residual CNN*), and the motion-free parameter maps as reference.

| | | 1.5T | | | 3T | | |
|---|---|---|---|---|---|---|---|
| **Correction** | **Metrics** | T1 | T2 | PD | T1 | T2 | PD |
| No correction | | 0.48 | 0.38 | 0.48 | 0.68 | 0.55 | 0.44 |
| Navigator | CCC | 0.72 | 0.61 | 0.61 | 0.82 | 0.75 | 0.60 |
| Navigator + residual CNN | | **0.78** | **0.71** | **0.71** | **0.87** | **0.83** | **0.83** |
| No correction | | 0.51 | 0.38 | 0.5 | 0.68 | 0.56 | 0.78 |
| Navigator | $R^2$ | 0.72 | 0.61 | 0.63 | 0.81 | 0.76 | 0.87 |
| Navigator + residual CNN | | **0.79** | **0.72** | **0.76** | **0.87** | **0.84** | **0.91** |

## A.3. Generalization analysis on clinical quantitative MRI

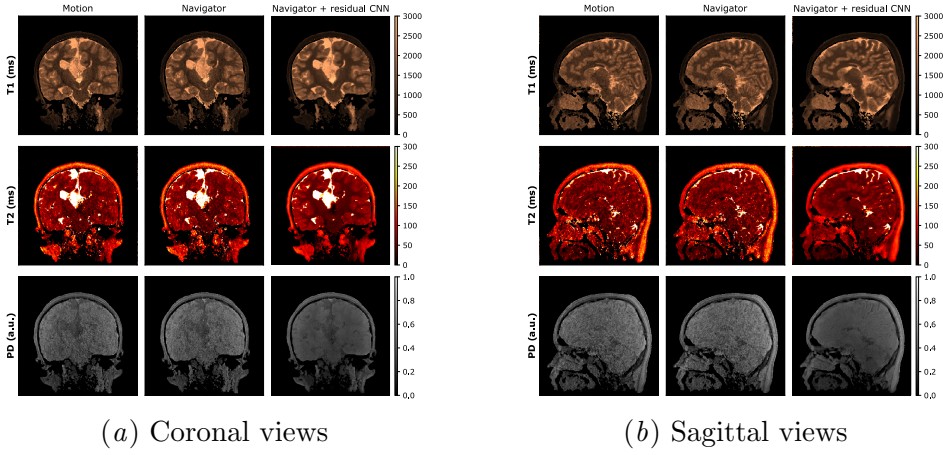

(*a*) Coronal views          (*b*) Sagittal views

Figure A.4: Proposed motion correction for the pediatric case acquired at 1.5T (coronal and sagittal views).

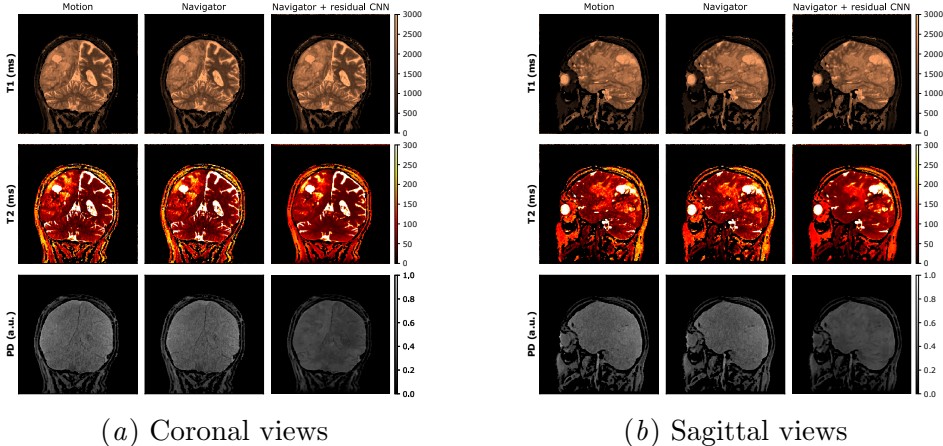

($a$) Coronal views $\qquad\qquad$ ($b$) Sagittal views

Figure A.5: Proposed motion correction for the adult patient's dataset acquired at 3T (coronal and sagittal views).

## A.4. Ablation study

Table A.2: Cross-validation experiment for quantitative comparison of the proposed *multiscale CNN* with global and local paths and a *singlescale CNN* comprising only two local paths, both applied after initial *Navigator*-based correction. Motion-correction performance is again summarized by concordance correlation coefficient (CCC) and coefficient of determination ($R^2$) between the motion-corrected and the motion-free parameter maps.

| CNN implementation | Metrics | 1.5T | | | 3T | | |
|---|---|---|---|---|---|---|---|
| | | T1 | T2 | PD | T1 | T2 | PD |
| Navigator + multiscale CNN (global + local path) | CCC | **0.78** | **0.71** | **0.71** | **0.87** | **0.83** | **0.83** |
| Navigator + singlescale CNN (2 local paths) | | 0.75 | 0.65 | 0.63 | 0.85 | 0.78 | 0.77 |
| Navigator + multiscale CNN (global + local pathway) | $R^2$ | **0.79** | **0.72** | **0.76** | **0.87** | **0.84** | **0.91** |
| Navigator + singlescale CNN (2 local paths) | | 0.75 | 0.66 | 0.68 | 0.86 | 0.8 | 0.87 |

