# OpenReview forum: "Residual learning for 3D motion corrected quantitative MRI: Robust clinical T1, T2 and proton density mapping"
_MIDL.io/2021/Conference — MIDL 2021_

### Official Review · AnonReviewer4 · 2021-03-04

**Confidence:** 4
**Preliminary Rating:** 3
**Recommendation:** Oral, Poster
**Final Rating:** 4

**Summary:**

The paper proposes a CNN-based method to correct for motion in parametric maps acquired using quantitative transient-state imaging. The motion is first corrected using a navigator method and the obtained maps are fed to a dual-pathway CNN (similar to DeepMedic), which predicts the residual motion. The proposed method clearly improves the quality of the parametric maps compared to navigator-based motion correction.

**Strengths:**

Very well-written and understandable paper on motion correction in qMRI. The study seems to be well executed.
-	The method is well described and illustrated (Fig 1)
-	The proposed simulation of motion data is an advantage of the method alleviating the need for paired motion corrupted and motion-free data
-	Experiments on healthy volunteers nicely show the capabilities of the method with convincing results. Two field strengths were used. Very good experimental design with in vivo data.
-	Two patients were scanned to show the generalization of the proposed approach. Results are convincing.
-	Good discussion of the results, as well as of the limitations and outlook

**Weaknesses:**

-	Method: What is the motivation to use fully-connected layers (FC)? In Fig. 1, the FCs have 3 outputs, so the CNN only predicts the center voxel of the patch (but in 2.3 it is written non-overlapping patches)? Please clarify. In the case that input and output are of the same dimension, were no border effects observed due to “same” convolutions in the local path?
-	Method: An ablation of the global pathway would help to better understand the influence of the provided context.
-	Validation: I do not completely understand the validation setup as two validations are described. The first is a 7/2/1 validation and the other a 10-fold cross-validation. Was the 7/2/1 repeated ten times? A clarification of how the network is tuned, and how the results are generated is needed.
-	Evaluation: I am not sure if a voxel-wise CCC and R2 are valid because there is a connection between neighboring voxels, i.e., the data is not independent. Also, does this make sense when a co-registration is involved in the evaluation? Maybe regions of interest might be more appropriate?
-	Method is not reproducible as no public code is mentioned in the paper


**Deanonymize Review:**

no

**Detailed Comments:**

-	The parametric maps in the figures seem to be skull-stripped. However, such a pre-processing is not mentioned in the paper. Was this only for visualization purposes?
-	For future work, a sensitivity analysis of the correction to the extent of simulated motion would be nice.

**Final Rating Justification:**

The authors addressed most of my concerns and, therefore, I revise my rating to strong accept.

**Justification Of The Preliminary Rating:**

Well executed and presented study proposing a method for motion correction in qMRI with convincing results. Therefore, I recommend the acceptance of the paper at MIDL. Minor modifications would further improve the clarity of the study.

**Paper Type:**

both

**Questions To Address In The Rebuttal:**

Please clarify the questions regarding the method and validation.

**Special Issue:**

no

---

> ### Author Response · Authors · 2021-03-16
> **Response to reviewer 4**
>
> We thank the reviewer for the positive and insightful comments, which we address below.
>
> 1.	Motivation and implementation of CNN architecture
>
> Thank you for making us aware of the potentially misleading illustration in Figure 1.
> In fact, the CNN preserves the spatial dimensions of the input patches throughout the network, i.e. also within the fully-connected layers. To do so, the fully-connected layer is implemented as a convolutional layer with 1×1×1 kernel sizes. We changed the illustration of the fully connected layers to describe the CNN implementation more clearly in Figure 1. We also clarified the description of the CNN architecture in the methods part accordingly.
> We rely on the fully-connected layer block as it allows us to handle arbitrarily large spatial relationships, comprised by the local and global paths, in a very efficient way.
> Thank you for your commenting on the patch-based processing. Apparently, our data processing routine at test time was not described precisely. In fact, we consecutively shift the parcellation of the 3D parametric maps into 3D patches of size 24×24×24 voxels along all directions. After patch-wise processing by the CNN, predictions are averaged to obtain the final output of the entire 3D volume, i.e. 200×200×200 matrix size. We changed the wording accordingly and now explain the actual processing in more detail.
>
> 2.	Ablation study
>
> In fact, an ablation study would be very interesting and insightful. However, we decided to concentrate on presenting a proof-of-concept demonstrating the potential of a CNN-based method for motion correction in fast 3D multiparametric MRI. Given the space limit, we focused on the most relevant aspects of our work regarding both the methodology and its evaluation on both healthy volunteers and patients.
> In the appendix of the revised manuscript we now include an additional table, where we compare motion correction performance of the proposed multiscale CNN with a singlescale CNN, with the same number of learnable parameters but two local pathways, each with prior navigator-based correction. Quantitative comparison indicates the impact of considering global contextual features. For both 1.5T and 3.0T field strengths, the improvement in CCC and R2 is more pronounced for T2 and PD, which seem to be more affected by remaining phase inconsistencies than T1.
>
> 3.	Experimental setup of cross validation
>
> Thank you for pointing out this potential misunderstanding. Yes, we use the data of 7 subjects for training, 2 for validation and the 1 held-back for testing. This experiment setup is repeated ten times so that all ten subject’s datasets are used for testing. We changed the respective paragraph accordingly.
>
> 4.	Performance evaluation
>
> Thank you for this comment. Indeed, there might be more sophisticated measures to better consider spatial neighborhood effects when evaluating parametric maps before and after correction. However, in this initial study we focused on the wide-spread and established CCC and R2 metrics within the brain region.
> Yes, voxel-wise metrics are sensitive to co-registration effects. As the entire quantitative analysis is based on the same registration, i.e. no-motion to motion space, comparing the two correction approaches relative to the non-corrected case is still valid.
>
> 5.	Open access policy
>
> Our CNN implementation and training routines are available on https://github.com/CarolinMA/MRP_MoCo. Navigator-based correction and motion simulation are incorporated into the reconstruction pipeline, the so-called recon_q package. Both the recon_q package and volunteer data can be made available by the research group at Imago7 upon reasonable request. Patient data are not publicly available due to restrictions on the use of confidential data in the written consent provided by participants.
>
> Detailed Comments:
>
> 1.	Use of skull stripping masks
>
> Exactly, brain masks were only used for visualization and quantitative evaluation.
>
> 2.	Sensitivity analysis of the correction to the extent of simulated motion.
>
> Thank you for this suggestion. That is indeed what we plan to do.

---

### Official Review · AnonReviewer2 · 2021-03-05

**Confidence:** 2
**Preliminary Rating:** 4
**Recommendation:** Oral
**Final Rating:** 4

**Summary:**

The authors propose a CNN- based method to motion-correct 3D quantitative MRI. The method is tested in simulated and real data, making very interesting use of physics inspired artificial motion corruption. The paper is well written and clear, and results present strong evidence of improvement over SOTA. The clinical application is clear and relevant.


**Strengths:**

* A solid methodology, clearly described. Well written paper.
* Good use of physics inspired artificial noise corruption to reduce the need for real data and to have a ground truth to compare with. Methods demonstrated also in in real volunteer and patient data, both adult and pediatric.
* Impressive results

**Weaknesses:**

* For a motion-correction method, there is little insight on how the motion was corrected. In other words, the network "magically" produces motion free images, but we do not know how the moved samples were rearranged in space. This black box effect, which is common to many DL approaches, has a number of problems in real clinical practice where an artefact due to abnormal anatomy or function could be corrected for, interpreted by the network as noise, for less common diseases that are not reflected by the training set. Can authors include a reflection on this?

* Figures feel a bit "all over the place". It makes the reader have to jump back and forth. I'd suggest to adjust the figures so that they are closer t o the sections where they are fist referenced.

**Deanonymize Review:**

no

**Detailed Comments:**

"We show that the 3D multiscale CNN naturally captures the 3D characteristics of both
the subject movements and the MR acquisition scheme" I fail to understand when they mean by "naturally". Please clarify.

Please include the meaning of the acronyms in the table captions

**Final Rating Justification:**

Authors have successfully addressed all my comments and I already thought this paper was worth accepting.

**Justification Of The Preliminary Rating:**

I don't see any major flaws in this paper, it is well written, well within the scope of MIDL, and presents a sound methodology and very appealing results. I can only recommend acceptance for presentation at the conference.

**Paper Type:**

both

**Questions To Address In The Rebuttal:**

I think that the points raised in the weakness section are fully addressable in the rebuttal.

**Special Issue:**

no

---

> ### Author Response · Authors · 2021-03-16
> **Response to reviewer 2**
>
> Thank you for the positive feedback and the opportunity to comment on your remaining questions.
>
> 1.	Insights on how motion is corrected
>
> We fully agree with the reviewer that it is crucial that critical diagnostic information is captured throughout the entire reconstruction pipeline. Particularly in the medical field, transparency is therefore indeed a major factor for potential integration into clinical routine. The navigator-based realignment outputs the estimated and subsequently corrected displacements. With the residual learning approach, the CNN learns to identify and resolve the remaining phase inconsistencies due to inter-segment motion that is not resolved. This way, the CNN is not exposed to the actual anatomical context, but particularly pointed to the sparse presentation of residual errors. As the motion patterns observed in neuro applications are rigid, the resulting phase-errors and artifacts are constant over the whole image volume, i.e. decoupled from the individual anatomy.
> As we demonstrate based on the pediatric and the adult patient cases, the CNN, which had not been trained on solely healthy volunteer data, nevertheless preserves disease-specific anatomical abnormalities. Building on this initial proof-of-concept, we plan a more extensive clinical evaluation of the proposed motion correction in the future.
>
> 2.	Arrangement of figures in the manuscript
>
> Thank you for pointing out these difficulties. We followed your advice and placed the figures closer to their reference in the manuscript. With the given page limit, we can only show one representative view and include the other views in the appendix.
>
> Detailed Comments:
>
> 1.	Clarification "We show that the 3D multiscale CNN naturally captures the 3D characteristics of both the subject movements and the MR acquisition scheme"
>
> The proposed 3D CNN allows us to capture both the motion patterns, which are in the 3D space, and the 3D relationships that arise from the 3D QTI acquisition.
> As suggested by reviewer 1, we shortened the abstract and removed this potentially misleading formulation.
>
> 2.	Acronyms in table captions
>
> We followed your suggestion and included the acronyms in the table caption.

---

> > ### Comment · AnonReviewer2 · 2021-03-18
> > **Response to rebuttal**
> >
> > Thanks for the answers.

---

### Official Review · AnonReviewer1 · 2021-03-07

**Confidence:** 4
**Preliminary Rating:** 4
**Recommendation:** Oral, Poster

**Summary:**

The authors propose to use a  3D multiscale convolutional neural network for retrospective motion correction for rapid 3D whole-brain multiparametric MRI acquisitions based on Quantitative Transient-state Imaging (QTI).
The mapping is learned from a physics-informed simulation to apply continuous motion-patterns to motion-free data and circumvent the necessity of large amounts of paired datasets.
Evaluation has been done on healthy volunteers and retrospective pediatric patients.

**Strengths:**

The authors propose a 3D multiscale residual CNN to learn the non-linear relationship between the motion-corrupted T1, T2 and PD maps and the residual error maps. This together with the use of a a physics-informed simulation framework for training are good ideas. One of these kinds where one wonders why this hasn't been tried yet?
Evaluation is good given the limited available resources. The method has been tested on prospectively acquired and retrospectively collected data.

**Weaknesses:**

Indeed using CNNs for motion correction has been explored in literature, for other imaging mechanisms though. It would be nice to see a brief discussion of CNN-based motion compensation methods for those beyond QTI, e.g. as explored since 2018 in ISMRM,
https://www.researchgate.net/profile/Kamlesh-Pawar-2/publication/327305084_Motion_Correction_in_MRI_using_Deep_Convolutional_Neural_Network/links/5b876fdb92851c1e123b336f/Motion-Correction-in-MRI-using-Deep-Convolutional-Neural-Network.pdf, or
Haskell MW, Cauley SF, Bilgic B, Hossbach J, Splitthoff DN, Pfeuffer J, Setsompop K, Wald LL. Network Accelerated Motion Estimation and Reduction (NAMER): Convolutional neural network guided retrospective motion correction using a separable motion model. Magnetic resonance in medicine. 2019 Oct;82(4):1452-61.
or the approaches that cannot rely on uniform motion corruption,
Miao S, Wang ZJ, Liao R. A CNN regression approach for real-time 2D/3D registration. IEEE transactions on medical imaging. 2016 Jan 26;35(5):1352-63.
Hou B, Khanal B, Alansary A, McDonagh S, Davidson A, Rutherford M, Hajnal JV, Rueckert D, Glocker B, Kainz B. 3-D reconstruction in canonical co-ordinate space from arbitrarily oriented 2-D images. IEEE transactions on medical imaging. 2018 Feb 19;37(8):1737-50.
Hou B, Miolane N, Khanal B, Lee MC, Alansary A, McDonagh S, Hajnal JV, Rueckert D, Glocker B, Kainz B. Computing CNN loss and gradients for pose estimation with Riemannian geometry. InInternational Conference on Medical Image Computing and Computer-Assisted Intervention 2018 Sep 16 (pp. 756-764). Springer, Cham.

The paediatric and adult patients in Fig 3 don't seem to show a lot of motion corruption. The CNN seems to mainly smooth/denoise the image.
Using a Navigator sequence seems to be the dominant factor for the observed improvements. How would this method perform if Navigators aren't available, e.g. severe motion or see refs above?

The dataset size is somehow hidden on page 3, ten healthy volunteers and two real patients. 'MR acquisition and reconstruction' should start with: 'In-vivo data from ten healthy volunteers...' or similar

**Deanonymize Review:**

no

**Detailed Comments:**

The abstract is too long.

**Justification Of The Preliminary Rating:**

I opt for strong accept because a) there's nothing really wrong with the paper except the few minor comments above, mainly looking a bit more beyond the scope of QTI and Navigator sequences in the background sections and b) 'weak' scores only introduce more randomness into the reviewing process.

**Paper Type:**

validation/application paper

**Questions To Address In The Rebuttal:**

How would the gain in reconstruction quality affect clinical downstream tasks, e.g. automated lesion segmentation, compared to using the Navigator on its own?

**Special Issue:**

no

---

> ### Author Response · Authors · 2021-03-16
> **Response to reviewer 1**
>
> Thank you for the positive feedback. In the following, we would like to respond to your comments.
>
> 1.	CNN-based methods applied for motion correction
>
> Thank you for drawing attention to these interesting works. We fully agree with you that a more detailed literature review on (CNN-based) motion correction helps to better understand the multifaceted problem as we also reflect on in our reply to reviewer 3. The broad spectrum of what kind of movements are to be corrected for and the therefore conceptionally different solutions are nicely reflected by the works mentioned by you: Pawar et al. and Haskell et al. present CNN-based methods to resolve motion-induced artifacts in 2D contrast-weighted brain MRIs. In contrast, Miao et al. demonstrate a CNN-based approach for pose estimation, i.e. to regress the parameters describing the underlying movements. Hou et al. approach the problem of orienting 2D image slices into a 3D atlas coordinate system to resolve motion between individual 2D slice acquisitions, e.g. for fetal brain imaging, also by regressing motion parameters.
> Independent of the actual application, they have in common that they are based on fully-sampled (Cartesian) MRI data or other (medical) imaging modalities.
> In case of 3D QTI, the massive undersampling (with an undersampling factor of 628) is what complicates motion estimation and  correction despite the high temporal resolution, which is generally beneficial for motion correction.
> Following your suggestion, we extended our literature review in the introduction to present a broader picture of (DL-based) motion correction, going beyond our demonstrated application for fast quantitative MRI based on transient-state acquisitions.
>
> 2.	Effectiveness of CNN with/without prior navigator-based correction
>
> We fully agree with you that the navigator-based approach has proven powerful in correcting for motion on a larger timescale as quantitatively reflected by the CCC and R2 improvements for the volunteer study. The navigators are reconstructed by taking advantage of the key-feature of 3D QTI, i.e. the temporally resolved k-space acquisition. As such, we don’t rely on a separate navigator sequence as the navigator images are readily available from every 3D QTI scan. We believe that our proposed method is that effective as it combines the strengths of both the navigator-based realignment and the subsequent alleviation of remaining artifacts due to motion on a smaller timescale by means of a CNN.
> Indeed, the patient cases don’t show as pronounced artifacts as the volunteer data. The milder artifacts are due to rather continuous than very abrupt and pronounced movements during the patient scans.
> Also, spiral readouts (as used in the underlying 3D QTI acquisition) do not have unique frequency- and phase-encoding directions, subject motion therefore causes rather diffuse image blurring instead of clearly visible ghosting artifacts as in case of Cartesian sampling. That is, slight up to moderate motion patterns, as suspected in case of the patient data, cause image blurring that is even more difficult to identify visually when it interferes with aliasing artifacts due to the massively undersampled k-space.
> While we demonstrate a proof-of-concept of this two-stage correction, we plan to further investigate how robust the motion correction is towards the actual motion parameters, such as timescale and range – as also suggested by reviewer 3.
>
> 3.	Information on dataset size
>
> We present dataset size and composition more clearly now.
>
> Detailed Comments:
>
> 1.	The abstract is too long.
>
> We shortened the abstract to present the work in a more compact way.
>
> Questions To Address In The Rebuttal:
>
> 1.	How would the gain in reconstruction quality affect clinical downstream tasks?
>
> Thank you for raising this point. Going beyond the quantitative parameter information itself, proton density, T1 and T2 maps obtained from 3D QTI can be used to synthesize image contrasts that mimic those of clinical sequences, such as T1-FSPGR, T2-FLAIR, Fast Spin Echo, MP2RAGE, etc. [1,2], by using the signal equations that describe each of these sequences. The preliminary patient data indicate that our proposed DL-method is promising, in that it enables the generation of sharper, less-artifacted parameter maps and therefore synthetic images (especially the T2-weighted ones) not only for radiological examination, but also in clinical downstream pipelines where conventional contrasts are used as input. We will address this topic in a dedicated, upcoming clinical evaluation.
>
> References:
>
> 1. Hagiwara A, Warntjes M, Hori M, et al (2017) SyMRI of the Brain: Rapid Quantification of Relaxation Rates and Proton Density, With Synthetic MRI, Automatic Brain Segmentation, and Myelin Measurement. Invest Radiol 52:647–657.
> 2. Gómez PA, Cencini M, Golbabaee M, et al (2020) Rapid three-dimensional multiparametric MRI with quantitative transient-state imaging. Scientific Reports 10:13769.

---

### Official Review · AnonReviewer3 · 2021-03-07

**Confidence:** 4
**Preliminary Rating:** 3
**Recommendation:** Poster

**Summary:**

A 3D CNN is trained on real data with artificially introduced motion to predict residual artifact maps. The network is applied after initial motion compensation via a navigator-based method. In order to investigate the generalisation capabilities of the presented approach, testing is performed on MRI images with real motion.

**Strengths:**

+ The paper is nicely written and well structured
+ I really like that you also introduce the navigator-based approach. What type of registration is used here?
+ Figure 1b nicely summarizes the deep-learning-based approach. I am not sure whether Figure 1a is really required.


**Weaknesses:**

- The amount of training and testing data is quite limited. However, you nicely deal with this issue by the motion simulation.
- Lacking comparison to other motion compensation approaches (except for the navigator-based approach)
- Please already mention in the abstract, that the deep-learning-based motion compensation is performed after initial navigator-based artifact reduction.


**Deanonymize Review:**

no

**Detailed Comments:**

- I was wondering, how robust the CNN works if the navigator-based artifact reduction fails. I would really like to see some results of the deep-learning-based motion compensation without initial navigator-based artifact reduction.
- In the caption of Figure 3, there is a "(b)" too much which should be removed

**Justification Of The Preliminary Rating:**

I recomment accepting the paper for MIDL 2021. It introduces a deep-learning-based motion artifact reduction approach for MRI images in a clean and well structured way. Experiments on clinical data with real motion show promising results.

**Paper Type:**

both

**Questions To Address In The Rebuttal:**

- How many trainable parameters does you network have? It seems a lot due to the fully-connected part in the end.



**Special Issue:**

no

---

> ### Author Response · Authors · 2021-03-16
> **Response to reviewer 3**
>
> Thank you for the positive and very comprehensive feedback and the opportunity to address open questions.
>
> 1.	Registration method used for navigator-based realignment
>
> For co-registering the navigator images to the baseline navigator, we used the normalized correlation coefficient (SPM12 software [1]).
>
> 2.	Figure 1a
>
> It is right that the content of Figure 1a is also contained in Figure 1b. Following your suggestion, we removed Figure 1a.
>
> 3.	Motion simulation framework as a means to deal with limited amount of training data
>
> We fully agree that availability of training data generally is the bottleneck for supervised learning in medical imaging. This was the driving force for the physics-informed motion simulation as presented in this study. The simulation framework does not only allow us to circumvent the need of paired, i.e. motion – no motion case data, but also enables a self-contained, well defined training environment with evidence-based motion ranges.
>
> 4.	Comparison to other motion compensation approaches
>
> Indeed, patient motion is a driving problem in clinical MR imaging with a number of conceptionally different correction methods, ranging from prospective, e.g. tracker-based approaches, to retrospective, image-based processing. As the appearance of motion artifacts highly depends on the individual acquisition, e.g. what readout schemes are used, the targeted clinical question, the condition of the respective patient and the body region to be imaged, there is no universal solution and motion compensation remains an open problem. [2, 3]
> Here, we demonstrate a retrospective motion correction method based on inherent navigator information that is readily available. This has the benefit that the acquisition does not have to be interrupted or adjusted, ensuring a short as possible scanning duration, without the need for separate navigator sequences or external tracking devices.
> As also suggested by reviewer 1, the introduction now includes a broader reflection on motion correction in general and DL-based methods in particular.
>
> 5.	Clarification already in abstract: the deep-learning-based motion compensation is performed after initial navigator-based artifact reduction
>
> Thank you for pointing this out. Following the suggestion of you and reviewer 1, we rephrased and shortened the abstract to make it clearer and more compact.
>
> Detailed Comments:
>
> 1.	Robustness of the CNN without prior navigator-based correction
>
> Thank you for this interesting suggestion.
> As we explain in the manuscript, the navigator-based correction improves image quality of the volunteer data more than for the patient cases. We attribute this to the fact that the involuntary head movements of the patients were less pronounced and rather continuous compared to the volunteers who were asked to intentionally move during scanning. As such, the patient cases illustrate how motion on a faster scale of what can be resolved by the self-navigators, is not sufficiently corrected by the self-navigator-based realignment. In line with your suggestion, we state in the outlook paragraph that we plan to investigate what motion scales can be resolved by the residual CNN. However, we emphasize that this is out of scope of this work.
>
> 2.	Caption of Figure 3
>
> Thank you for pointing this out. We removed the “(b)” in the caption.
>
> 3.	Number of trainable parameters
>
> The CNN has about 2.3 million parameters in total. The fully-connected block contributes about 80,000 and allows us to efficiently combine all the information captured within the local and global path. Corresponding training time was about 1 hour on a NVIDIA Titan Xp gpu.
>
> References
>
> 1. Penny W, Friston K, Kiebel S, Nichols T (2006) Statistical Parametric Mapping: The Analysis of Functional Brain Images. Academic Press, Massachusetts
> 2. Godenschweger F, Kägebein U, Stucht D, et al (2016) Motion correction in MRI of the brain. Phys Med Biol 61:R32-56.
> 3. Zaitsev M, Maclaren Julian, Herbst M (2015) Motion Artefacts in MRI: a Complex Problem with Many Partial Solutions. J Magn Reson Imaging 42:887–901.

---

### Meta-Review · Area_Chair1 · 2021-03-26

**Recommendation:** Accept (Oral)

**Metareview:**

All reviewers agree that this is a well-written paper describing a neat application. I think it would make a good oral.

**Paper Type:**

validation/application paper

---

### Decision · Program_Chairs · 2021-03-31

**Decision:**

Accept

**Comment:**

Congratulations your paper has been selected as a long oral.